# Emerging Potential of Metabolomics in Thyroid Cancer—A Comprehensive Review

**DOI:** 10.3390/cancers17061017

**Published:** 2025-03-18

**Authors:** Sonam Kumari, Andrew Makarewicz, Joanna Klubo-Gwiezdzinska

**Affiliations:** Metabolic Diseases Branch, National Institute of Diabetes and Digestive and Kidney Diseases, National Institutes of Health, Bethesda, MD 20892, USA; sonam.kumari@nih.gov (S.K.); andrew.makarewicz@nih.gov (A.M.)

**Keywords:** thyroid cancer, metabolomics, biomarkers, mass spectrometry, resistance

## Abstract

Thyroid cancer is the most frequent malignant endocrine disorder, with a high occurrence in the female population. Cancers rely on certain metabolic pathways to secure their growth and metastatic potential. Therefore, cancer metabolomics may serve as a novel tool in precision medicine focused on thyroid cancer diagnosis, prognosis, and prediction of its responsiveness to standard therapy. In this review, we summarize the various applications of metabolomics in thyroid cancer research. This includes areas such as biomarker discovery and environmental risk factors of thyroid cancer, non-invasive thyroid cancer diagnostics, disease subtyping, markers of therapy resistance, and radiotoxicity. This evolving technique may be useful in the coming years to effectively differentiate between thyroid cancer characterized by high and low aggressiveness and to help monitor targeted therapies.

## 1. Introduction

Thyroid cancer is the most prevalent endocrine malignancy [1,2,3]. The occurrence of thyroid nodules is attributed to a combination of various factors such as genetics, environment, sex, and age. Thyroid nodules mostly occur in females and the incidences rise with age [4,5]. The rise in incidence in thyroid cancer has been attributed, to a large extent, to over-detection due to excessive imaging that was observed in the early 2000s, and that has plateaued in recent years [6] as more conservative diagnostic criteria have been applied [7].

Thyroid cancer originates from either the follicular cells or the parafollicular cells of the thyroid gland. Follicular cells are mainly involved in the uptake of iodine and production of thyroid hormones, and lead to the majority of thyroid carcinomas. On the other hand, parafollicular neuroendocrine cells are associated with the secretion of calcitonin, which plays a role in calcium metabolism. Histologically, thyroid cancer is divided into the following four groups: papillary, follicular, anaplastic (all derived from follicular epithelial cells) and medullary (originating from parafollicular cells). Papillary thyroid cancer (PTC) accounts for about 85%, follicular thyroid cancer (FTC) for almost 7–14%, medullary thyroid cancer (MTC) for about 3%, and anaplastic thyroid cancer (ATC) for approximately 2% of thyroid cancer incidences [8].

The most commonly used method to diagnose thyroid cancer is FNAB (fine-needle aspiration biopsy) with a cytological analysis [9]. The Bethesda System classifies FNAB results into the following six categories: (I) nondiagnostic, (II) benign, (III) atypia of undetermined significance, (IV) follicular neoplasm, (V) suspicious for malignancy, and (VI) malignant [10,11]. However, about 30% of FNABs are unable to distinguish benign from malignant owing to their indeterminate character (Bethesda III/IV) [12]. This uncertainty leads to the repetition and re-evaluation of FNABs or the application of molecular testing, which increases the diagnostic accuracy [13,14]. Despite advancements in molecular testing, unnecessary thyroidectomies or lobectomies are still prevalent, having been reported for 56–68% of nodules with Bethesda III/IV cytology and 21% with Bethesda V cytology [15,16]. Moreover, molecular tests commercially available in the U.S. (Afirma GSC, Thyroseq, and ThyGeNEXT/ThyraMIR) are not widely used in Europe and Canada due to high costs and reimbursement issues [17,18,19]. Additionally, the processing time for the molecular testing of indeterminate thyroid nodules takes around 2–4 weeks [17]. Therefore, there is a dire need for more cost-effective and rapid diagnostic techniques.

The use of “OMICS” technology, which is composed of transcriptomics, genomics, metabolomics, and proteomics, is very attractive in the field of cancer. In particular, metabolomics is efficient at providing information about the physiological condition of a biological specimen and its phenotype [20]. In general, this technology targets metabolites < 1500 Da in molecular weight [21]. It has been found to be useful for investigating the metabolism of cancer cells and finding new diagnostic and prognostic markers [22]. A number of studies have already reported the identification of biomarkers for breast [23], prostate [24], kidney [25], and brain [26] cancer using metabolomics analysis. The use of metabolomics can help in the investigation of distinct phenotypes resulting from environmental or genetic factors [27]. Metabolic pathways can be correlated with cancer biomarkers because cancer development and progression are associated with metabolic switches that could be reflected in metabolic profiles. Accumulating evidence suggests that there are marked changes in normal and malignant thyroid tissue samples with respect to the level of lipids and their metabolites [21,28,29,30,31,32]. Lipids have a significant role in cellular metabolism and growth and are, therefore, associated with oncogenic pathways [33,34].

This review summarizes the recent developments in the field of metabolomics utilization in thyroid cancer and outlines the potential applications of this promising tool.

## 2. Search Strategy

We conducted a comprehensive literature review utilizing the PubMed search engine. Data were analyzed from full-length original articles, case studies, and clinical reports published between 1 January 1980 and 31 December 2024. Keywords included “thyroid cancer”, “thyroid carcinoma”, “metabolomics”, “targeted metabolomics”, “untargeted metabolomics”, “biomarkers”, “therapy resistance”, “mass spectroscopy”, “MRI spectroscopy”, “metabolism”, and “metabolites”. The search was restricted to manuscripts published in English, and primarily focused on articles based on their originality, relevance, and clinical significance. The search strategy is summarized in Figure 1.

## 3. Untargeted and Targeted Metabolomics

Although there are several studies utilizing transcriptomics, genomics, and proteomics analyses in the area of thyroid cancer [35,36,37], metabolomics has not been extensively employed. Metabolomics can be classified into untargeted and targeted metabolomics [38], as outlined in Figure 2.

The two most important techniques used for the analysis of thyroid cancer are matrix-assisted laser desorption/ionization–mass spectrometry imaging (MALDI-MSI) and high-resolution magic angle spinning nuclear magnetic resonance (HR-MAS NMR) [39,40]. Marked differences were found between normal and tumor tissues in these studies. Owing to the diversity of metabolites with respect to their composition and physicochemical attributes, a unique analytical method is not available for the complete metabolome. Also, the various methods used differ with respect to their sensitivity and specificity [21].

NMR spectroscopy and mass spectroscopy, in combination with gas chromatography (GC) or liquid chromatography (LC), are frequently used to evaluate the metabolic profiles of biological samples. The differences between these routinely used analytical techniques are summarized in Figure 3. It is extremely important that the sample is adequately collected, stored, and processed for the metabolomics analysis. Some samples (for example, urine) can be directly evaluated. In some instances, urease is required to be added to the urine samples to remove urea, which could interfere with the quality of the GC assay [41,42]. In a GC/MS analysis, the method of derivatization plays a significant role in converting the polar metabolites into volatile compounds. On the other hand, NMR requires balancing the pH using a phosphate buffer. Also, if the samples require further dilution, deuterium is used [43]. Deproteinization is essential for blood samples owing to the presence of abundant protein. The sample processing requires a volume of at least 100–500 µL of biofluids.

In general, tissue processing is more complicated as it contains blood and is associated with tissue heterogeneity. Often, saline is utilized for washing the samples, and techniques such as LCM (laser capture microdissection) are employed prior to storage. Moreover, in order to obtain metabolites from tissues, an extraction buffer and homogenizer are needed [44]. Thereafter, the specimens are dried and an appropriate solvent is added to them to proceed with LC/MS. In the case of GC/MS, further chemical-derivatization steps are required (Figure 3).

A crucial factor to consider for metabolomics assays is the use of suitable chemical solvents for the samples. Some solvents, including butanol/methanol (BUME), chloroform (methyl tert-butyl ether, or MTBE), or methanol, are used for GC/MS as well as LC/MS. For NMR studies, trimethylsilylpropanoic acid (TSP) or 4, 4-dimethyl-4-silapentane-1-sulfonic acid (DSS) are conventionally used as reference standards for appropriate metabolite quantifications [45]. Once the data are collected, normalization and mean centering are conducted to process the data. Univariate unsupervised multivariate statistics or supervised multivariate statistics can be used for the data analysis. There are several tools available, such as Ingenuity Pathway Analysis (IPA, by Qiagen, main headquarters in Hilden, Germany), Cytoscape (version 3.10.3), and Metaboanalyst 6.0, that can be utilized for the quantification of data. The various steps involved in this assay are summarized in Figure 4. Untargeted metabolomics focuses on the global analysis of known as well as unknown metabolites and is, therefore, unbiased. The data obtained can be further utilized for relative quantifications, and serve as diagnostic or prognostic tools.

On the other hand, targeted metabolomics can identify and analyze a limited number of pre-defined metabolites or confirm and validate the data obtained from unsupervised screening. For example, some of the metabolites that can be targeted have been consistently detected in malignancies, including thyroid cancer. Elevated lactate and changes in the levels of leucine, choline, phenylalanine, and tyrosine have been reported. Increased lactate can be attributed to the Warburg effect, which suggests that cancer cells demonstrate upregulated glycolysis with high lactate secretion, even under abundant oxygen conditions [46]. Lactate has an essential role in cell migration, angiogenesis, and self-metabolism. It has also been reported to be a potential biomarker in various cancers, including breast, pancreatic, lung, and thyroid cancer [47,48,49,50].

## 4. Metabolomics for Biomarker Discovery

Metabolomics has become a gateway for studying metabolic pathways, phenotypes and, metabolic reprogramming in thyroid cancer. Significant alterations have been documented in the thyroid cancer metabolism of glucose, amino acids, and lipids as well as mitochondrial respiration compared with non-malignant thyroid tissue [51,52]. Thyroid cancer research can capitalize on these metabolic alterations as an avenue for identifying new biomarkers. Metabolomics studies of thyroid cancer have commonly been based on NMR techniques, specifically, 1H-NMR metabolomics with an untargeted approach [53]. However, mass spectrometry has gained significance in recent work given the limitations of NMR, such as its lower sensitivity [52,54]. Given the enhanced sensitivity of mass-spectrometry-based metabolomics, a broader range of metabolites can be detected, providing a more comprehensive understanding of metabolic alterations in thyroid cancer [53]. Other metabolomics techniques, such as Fourier-transform infrared spectroscopy, are also gaining attention in thyroid cancer research [55]. Razavi et al. conducted a systematic review and meta-analysis of NMR-based metabolomics studies, identifying forty differential metabolites associated with thyroid cancer (Table 1) [56]. These metabolites could serve as promising biomarkers for early diagnosis. In a subsequent study, Razavi et al. identified potential biomarkers for thyroid cancer, including the commonly downregulated metabolite, myo-inositol [57]. Myo-inositol may help differentiate malignant tissue from healthy tissue, offering a non-invasive diagnostic tool using biofluids such as plasma [57]. The diagnostic performance measured using the area under the curve (AUC) of the metabolic profile, which included myo-inositol, was 85.3%, comparable with that of FNA (90%) [58,59]. In a recent study by Yu et al., a novel metabolic biomarker signature (17 biomarker panel) was identified to distinguish between PTC and benign thyroid nodules (BTNs) (Table 1). Interestingly, the diagnostic accuracy of this test was 92.72% in the validation cohort, which is comparable with the standard FNAB. However, while FNAB is a gold standard, the 30% prevalence of cytologically indeterminate thyroid nodules suggests that metabolomics may complement FNAB. Moreover, when combined with ultrasound, it might strengthen the diagnostic accuracy even more [60]. Metabolomics has been applied to indeterminate and/or nondiagnostic FNABs [55,61,62,63,64]. These studies demonstrate that metabolomics can effectively reclassify indeterminate FNAB specimens, with potential accuracy rates of up to 95%. Accurately classifying indeterminate nodules as malignant or benign is crucial for reducing the need for unnecessary diagnostic surgeries (Table 1).

Mass spectrometry imaging (MSI) of metabolites offers another approach for thyroid cancer diagnostics. Although the non-invasive analysis of metabolites can provide insightful quantifications, MSI allows the spatial mapping of metabolites. As tumors exhibit significant heterogeneity, metabolite spatial mapping gives a more complete picture of metabolic plasticity beyond the sole concentration [65,66]. In thyroid cancer, MSI has had many recent applications. MALDI-MSI has shown to be capable of identifying PTC from benign tissue through the lipid profiling of formalin-fixed tissue [67]. Lipid profiling revealed that phosphatidylcholines, sphingomyelins, and phosphatidic acids were more abundant in malignant tissue versus benign. A study utilizing desorption electrospray ionization mass spectrometry (DESI-MS) imaging provided a potential method for analyzing FNAB samples to reduce the number of unnecessary diagnostic surgeries [68]. The metabolic profiles of the FNAB samples were obtained using DESI-MS, classifying malignant PTC from benign tissue with an AUC of 0.98, higher than that of FNAB. Another recent study using MALDI imaging found that N-glycan abundance was much higher in normal thyroid tissue compared with cancer tissue [69]. Mao et al. used air-flow-assisted desorption electrospray ionization mass spectrometry imaging (AFAIDESI-MSI) to distinguish benign follicular thyroid adenoma (FTA) from malignant carcinoma based on the metabolic signature, finding both phospholipids and fatty acids to be more abundant in FTA than FTC [70]. Given this information, they also found the fatty acid synthase (FASN) and calcium-independent phospholipase A2 (iPLA2) metabolic enzymes to be significantly upregulated in FTA. These metabolic alterations had a diagnostic AUC of 73.6%, with the potential to classify borderline follicular tumors. Overall, mass spectrometry imaging enhances thyroid cancer diagnostics by providing spatial metabolic profiling that improves the understanding of tumor heterogeneity.

**Table 1 cancers-17-01017-t001:** Emerging Trends in Metabolomics Thyroid Cancer Research.

Reference	Review Design	Biospecimen Used	Significantly Altered Metabolites
Khatami et al., 2019 [71]	Systematic review of 31 metabolomic studies (15 targeted and 16 untargeted) investigating metabolite biomarkers of TC. All metabolomic techniques included in search criteria.	Plasma, serum, urine, or FNA specimens. Malignant TC vs. control (healthy, benign nodules, goiter)	Citrate ↓ Lactate ↑
Abooshahab et al., 2022 [53]	Systematic review of metabolomics in endocrine cancers. 35 articles published from 2010–2022 on thyroid cancer metabolomics. Techniques included NMR (15 papers), GC/MS (8 papers), and LC/MS (12 papers).	Tissue, serum/plasma, urine, FNA samples Malignant vs. benign tumors	Lactate ↑ Choline ↓ Mono- and disaccharides, and TCA intermediates altered
Coelho et al., 2020 [54]	Review includes 45 original studies on TC metabolomic biomarkers. NMR (21 papers), MS (19 papers), other techniques (5 papers). Spatial metabolomics applied in several listed studies.	Tissue, plasma, serum, urine, feces, breath TC vs. healthy/benign controls	Choline ↑ Lactate ↑ Tyrosine ↑
Abooshahab et al., 2024 [72]	Review of metabolomic studies on TC cell lines. 7 papers identified. MS (6 papers) and NMR (1 paper).	TC cell lines	Various alterations in glycolysis and TCA cycle metabolites
Neto et al., 2022 [55]	Review of studies using FTIR spectroscopy to characterize normal vs. tumor samples. 13 papers met the criteria.	Thyroid tissue and cytology samples	Lipids ↓ Carbohydrates ↓ Lipid metabolism ↑
Razavi et al., 2024 [56]	Systematic review and meta-analysis of NMR-based metabolomic studies. 12 studies met the search criteria.	Tissue and FNAB specimens. Malignant vs. benign.	Lactate ↑ Alanine ↑ Citrate ↓
Nagayama et al., 2022 [51]	Summarize the recent findings of metabolic reprogramming in TC as well as recent reports of metabolism-targeted therapies.	Thyroid tissue and cytology samples	Glucose metabolism ↑ Amino acid metabolism ↑ Lipid metabolism ↑

TC: thyroid cancer; NMR: nuclear magnetic resonance; FNA: fine-needle aspiration; MS: mass spectrometry. ↑ refers to upregulation, ↓ means downregulation.

Recent studies have also focused on developing thyroid cancer metabolic profiles using biofluids. This is a minimally invasive approach compared with FNA and/or surgery to collect a biospecimen. The study by Berinde et al. supported the potential of using metabolic biomarkers to discriminate between PTC, benign nodules, and healthy subjects [73]. Using HPLC-QTOF-ESI+-MS technology, they identified 17 metabolites, including lipid- and selenium-related molecules, that had a discriminatory value of 0.97 for differentiation between PTC and healthy control patients [73]. Another study by Berinde et al. investigated the urinary biomarkers used to differentiate PTC and benign nodules from healthy controls [74]. They identified ten classes of metabolites, mostly lipid-related, that served as diagnostic biomarkers, with an accuracy of 65–73%. This suboptimal accuracy suggests that urinalysis might be subject to a significant bias from several confounding excreting metabolites, preventing its use as a sole diagnostic tool.

Analyzing breath metabolites is another non-invasive approach for the early diagnosis of thyroid cancer [75]. Biomarkers were identified in both end-tidal and mixed expiratory breath, with area under the curve (AUC) values greater than 90% that effectively distinguished between PTC and healthy control groups. Potential classes of breath biomarkers identified in the study by Li et al. included alkanes, esters, ketones, and fluorine/chlorine organic compounds.

Additionally, Razavi et al. used 1H-NMR spectroscopy to profile the metabolites of PTC plasma samples [59]. Their profiling identified increased levels of leucine and lysine along with decreased levels of pyruvate and tyrosine in patients with PTC compared with healthy individuals [59]. Although the diagnostic accuracy was 85%, the sample size was relatively small, and the study requires further validation.

Zhang et al. also identified ten amino acid metabolites in saliva that were able to differentiate PTC from healthy controls [76]. These metabolites, used as individual biomarkers, had AUC values ranging from 67.8 to 83.3%. However, when alanine, valine, proline, and phenylalanine were combined into a diagnostic panel, the AUC improved to 93.6%, suggesting that combining multiple biomarkers could increase the accuracy of early diagnosis. Both urinary and serum metabolites were investigated by Chen et al., and they identified six metabolites with the potential of diagnosing PTC (AUC = 95.2% in samples from both females and males) [77]. To further validate this study and incorporate PTC staging into the metabolite analysis, a larger sample size is necessary as biofluids were collected from only 124 untreated PTC patients, 76 untreated BTN patients, and 116 healthy controls, respectively.

Importantly, when identifying these biomarkers, it is crucial to consider how they are influenced by confounding physiological factors. For example, iodine is closely linked to thyroid function and may affect different biomarkers of thyroid cancer [78]. Liu et al. found that iodine nutrition affects the diagnostic accuracy of several serum metabolites [78]. In addition, environmental chemical disruptors can play a significant role in the interpretation of metabolomics data. Wang et al. studied how exposure to per- and polyfluoroalkyl substances (PFASs) affected the serum metabolome and thyroid cancer risk [79]. Their study suggested that PFAS exposure may disrupt several free fatty acid metabolites involved in thyroid cancer pathogenesis [79]. Similarly, Song et al. examined the impact of polybrominated diphenyl ethers (PBDEs) on the metabolic signature of PTC [80]. Their findings highlighted octopamine and 5-hydroxyindole as potential diagnostic biomarkers for PTC [80].

To summarize, these findings suggest that metabolic biomarkers, such as those found in biofluids, combined with novel imaging techniques may significantly add to the armamentarium of thyroid cancer diagnosis and might serve as minimally invasive and affordable ways to diagnose thyroid cancer [81].

## 5. Metabolomics in Disease Subtyping

Recent studies have also focused on using metabolomics profiling to further phenotype thyroid cancer. Wojakowska et al. showed that the metabolic signature of classic PTC is different compared with follicular variant PTC as it is characterized by upregulated citric acid and downregulated gluconic acid [82]. These data may suggest that the main oncogenic drivers of classic PTC such as *BRAFV600E* are associated with different metabolic profiles from *RAS*-driven tumors that usually present a follicular pattern of growth [83]. Similarly, Kim et al. explored the metabolic phenotypes of thyroid cancer molecular subtypes. The key findings included FTC with *RAS* mutations having an enhanced tricarboxylic acid cycle and branched-chain amino acid degradation. Compared with PTC and FTC subtypes, ATC displayed higher upregulated one-carbon metabolism as well as pyrimidine metabolism [84]. Qu et al. aimed to characterize PTC patients based on the risk of recurrence [85]. Their multi-omics strategy included the metabolomics profiling of 503 metabolites from 102 tumor and 37 paired normal tissue samples [85]. Patients with a high risk of recurrence had upregulated levels of triglycerides, free fatty acids, histamine, and kynurenine. Additionally, the upregulation of metabolites in amino acid biosynthesis and glycolysis was found in patients with a high recurrence risk. These findings corroborate pre-clinical and clinical studies showing that molecular alterations in similar metabolic pathways are associated with poor PTC outcomes [86,87,88,89,90,91].

Lee et al. used liquid chromatography–mass spectrometry (LC-MS) to perform the metabolic profiling of 17 thyroid tumor–normal tissue pairs [92]. Their data showed an increase in serine and related metabolites in undifferentiated thyroid cancer, highlighting the role of the serine/glycine pathway in thyroid cancer cell proliferation/de-differentiation [84]. Similarly, Huang et al. performed a metabolite set enrichment analysis of thyroid cancer, identifying enriched pathways such as arachidonic acid and cysteine/methionine metabolism, which are both associated with the ferroptosis pathway [93]. Furthermore, they found that enzyme longevity assurance homologue 2 (LASS2) in association with the transferrin receptor (TFRC) could suppress metastasis in thyroid cancer by regulating ferroptosis, a critical pathway involved in tumor growth and proliferation [93].

Metastatic disease in thyroid cancer is associated with a poorer prognosis, and the early detection of the metastatic potential or the identification of the metabolic pathways that could be targeted to eradicate metastatic disease are of utmost importance. Xu et al. found 31 differential metabolites in serum from patients with lymph node metastasis compared with those without metastatic disease, including elevated levels of L-proline, L-tryptophan, and 5-hydroxylysine and decreased levels of deoxycholic acid, erucic acid, and glycerophosphocholine [94]. Another study identified increased lactate associated with lymph node metastasis [95]. Shen et al. explored the metabolic signature of PTC with distant metastasis, revealing 31 differential serum metabolites linked to glucose, amino acids, lipids, gut microbes, and vitamin metabolism [96]. Similarly, another study found six altered metabolites associated with metastatic thyroid cancer, including ascorbic acid, guanidinoacetic acid, betaine, pyruvate, phenylacetic acid, and adenosine [97]. Although these studies were limited by small sample sizes, they offer promising avenues to understand and target the metabolic shifts occurring in metastatic disease and potentially testing them as biomarkers of early metastatic disease. They may also potentially serve to capture the metabolic reprogramming in response to targeted therapy. By analyzing changes in metabolites, this technique provides a real-time picture of a drug’s anti-tumor effects at the metabolic level. Unlike traditional methods that focus on tumor imaging and biomarker measurements (such as thyroid-cancer-specific thyroglobulin or calcitonin), metabolomics can monitor the treatment response using the analysis of metabolic reprogramming. For example, a study by Ouyang et al. showed how metabolomics could be used to evaluate the therapeutic benefits of metformin in thyroid cancer [98]. By integrating metabolomics with transcriptomics, they identified significant molecular changes associated with the treatment. Specifically, they observed altered metabolites in glutathione metabolism and the TCA cycle as well as the inhibition of glycolysis. A study by Thakur et al. also revealed downregulated oxidative phosphorylation in response to metformin treatment [99]. These findings all point to the ability of metabolomics to assess metabolic shifts associated with treatment.

As resistance to targeted therapies is the main reason for thyroid-cancer-specific mortality, we next focus on the role of metabolic profiling in unresponsiveness to standard treatments.

## 6. Metabolomics for Overcoming Therapy Resistance

Resistance to radioiodine therapy continues to be a major cause of thyroid cancer mortality. The timely identification of radioiodine resistance in patients is crucial for selecting appropriate alternative treatment options, such as tyrosine kinase inhibitors (TKIs). Zheng et al. used LC-MS to identify metabolic differences between RAI-resistant and non-resistant thyroid cancer patients [100]. Their analysis revealed differences in the metabolites of the phenylalanine and tyrosine pathways, which could serve as biomarkers for predicting RAI resistance [100]. However, these biomarkers have not been prospectively tested to validate their utility in clinical practice. In a similar study, Wang et al. found that ketone body metabolism was significantly altered in radioiodine (RAI)-resistant differentiated thyroid cancer (DTC) patients [101]. Specifically, the upregulation of acetoacetate in non-RAI-resistant patients enhanced 131I uptake and increased sodium/iodide symporter (NIS) and thyroid-stimulating hormone receptor (TSHR) levels [101]. This key finding suggests potential therapeutic applications of inducing ketogenesis to overcome RAI resistance in DTC. Untargeted metabolomics performed by Yu et al. revealed that polyfluoroalkyl substance (PFAS) exposure may be associated with resistance to RAI therapy [102]. The authors identified the enrichment of perfluorodecanoic and perfluorononanoic acids in RAI-refractory DTC patients, which was associated with thyroid cancer progression and a reduced expression of NIS, which is necessary for RAI uptake [102].

Beyond RAI resistance, metabolomics can provide insights into RAI-therapy-induced radiotoxicity. Two studies by Lu et al. examined the effects of a 131I treatment on gut metabolites in DTC patients [103,104]. LC-MS/MS untargeted metabolomics found downregulated pathways of linoleic acid, arachidonic acid, and tryptophan metabolism after 131I therapy [103]. These radiation-sensitive pathways are suspected to play a crucial role in protecting against radiation toxicity, which could have long-term implications for DTC patients. Despite the downregulation of arachidonic-acid-related metabolites and pathways post-131I therapy, arachidonic acid supplementation in mice reduced radiotoxicity, offering the potential to mitigate the side effects of radiation therapy [104]. These findings highlight the importance of metabolomics profiling in understanding radiotoxicity and suggest potential therapeutic strategies for minimizing side effects.

As RAI-resistant thyroid cancer requires therapy with targeted TKIs, understanding resistance to such therapy is of vital importance. Resistance to single-agent chemotherapeutics remains a major challenge in managing aggressive thyroid cancer. Metabolomics has proven useful for identifying metabolic alterations associated with therapy resistance. A recent study by Kumari et al. utilizing the technique of unsupervised metabolomics revealed that thyroid cancer cell lines resistant to standard-care TKI lenvatinib clustered separately from non-resistant cell lines, suggesting a distinct metabolic profile [105]. Xu et al. investigated the synergistic effects of *BRAF* and *PIM1* inhibitors on *BRAFV600E*-positive PTC cell lines [106]. *PIM1* is an oncogenic serine/threonine kinase involved in the cell cycle, drug resistance, cell survival, and cell proliferation. The expression of *PIM1* is upregulated in *BRAFV600E*-mutated thyroid cancer, which could possibly serve as an escape mechanism for *BRAF* inhibitors. The metabolomics data revealed that the combination therapy limited the availability of essential amino acids (phenylalanine, arginine, and glutamate) for cell growth and proliferation while also downregulating nucleotide synthesis pathways [106]. Liu et al. integrated metabolomics and transcriptomics to uncover the mechanisms contributing to thyroid cancer resistance to anlotinib that was used in clinical trials for metastatic progressive thyroid cancer [107]. A combined “omics” analysis revealed a high expression of glutamate in anlotinib-resistant thyroid cancer cell lines [107]. Additionally, eight genes closely related to glutamate involvement in disease pathways were differentially expressed [107]. Of these genes, *LPAR1* (lysophosphatidic acid receptor 1) is suspected to be a potentially important target for overcoming resistance to anlotinib [107]. *LPAR1* plays a role in cell proliferation, migration, and survival by mediating the signaling pathways involved in tumor growth [108].

These findings emphasize the ability of metabolomics to detect metabolic shifts in resistant clones and identify targets for therapies aimed at overcoming drug resistance in thyroid cancer. Therefore, it is important to develop models investigating the signal detected by metabolomics studies in appropriate in vitro and in vivo models to understand potential causal relationships and associations with thyroid cancer.

## 7. Models for Metabolomics Studies

Model development is essential for advancing metabolite-based approaches for screening thyroid cancer biomarkers. Kuang et al. developed a machine learning model with the potential for screening for thyroid cancer using metabolite biomarkers [109]. Their machine learning model identified PTC-related metabolites with an accuracy of 87.3%. The combination of machine learning and artificial intelligence with metabolite biomarkers holds great promise for the rapid screening and accurate diagnosis of thyroid cancer [109].

Cell culture models allow for controlled and reproducible studies of the metabolic behavior of thyroid cancer [72]. Metabolic alterations have been documented using cell lines from multiple subtypes of thyroid cancer [72]. In the oncocytic thyroid cancer cell line XTC.UC1, a metabolomics analysis revealed that glycolytic intermediates were utilized not only for upregulated glycolysis, but also lipid metabolism and serine synthesis [110]. As this diversion of metabolites was not observed in non-oncocytic models, it was suspected that defective mitochondria-rich oncocytic cancer cells could be compensating for their impaired mitochondrial oxidative phosphorylation [110]. Kumari et al. described how oncocytic thyroid cancer is characterized by genetic alterations leading to decreased oxidative phosphorylation and increased glycolysis [111]. Another study utilized PTC-derived cell lines to investigate the cytotoxic effects of vitamin C, revealing that it induces oxidative stress and alters glycolysis and the TCA cycle, leading to cancer cell death and suggesting potential treatment strategies [112]. Additionally, Chen et al. proposed a method involving gold-doped zirconium-based metal–organic framework (ZrMOF/Au) nanostructures for the metabolic profiling of thyroid cancer [113]. Cristiani et al. also developed a 3D thyroid organoid model from human primary thyrocytes for an investigation of thyroid metabolism [114]. Their study revealed that thyroid-metabolism-related genes such as *TPO*, *TSHR*, *PAX8*, *TTF-1*, *NIS*, *IYD*, and *TG* maintained functionality in the 3D organoids, whereas 2D cultures lost the phenotype over time. Although thyroid-cancer-patient-derived organoids have been established [115,116], further advancements are needed to translate these models into use in metabolomics studies.

Together, these efforts highlight the growing potential of different metabolomics approaches for thyroid cancer.

## 8. Metabolomics: Bridging Other Omics

Metabolomics studies tend to be much more powerful when coupled with genomics, transcriptomics, and proteomics. Dhuli et al. reviewed how the integration of “OMICS” sciences has improved the diagnosis, prognosis, and treatment of thyroid cancer [117]. In particular, they found that major advancements in precision medicine have stemmed from combined efforts of “OMICS” disciplines. For example, genomics and transcriptomics have uncovered critical mutations driving thyroid cancer, whereas proteomics and metabolomics have identified biomarkers enhancing the accuracy of thyroid cancer diagnostics and prognostics.

Molecular markers discovered using multi-omics approaches have enabled the development of personalized therapies such as *NTRK* or *ALK* fusions to improve the overall prognosis for thyroid cancer patients [118]. Another comprehensive analysis displayed the complexity of each thyroid cancer subtype and underscored the translational benefits of a multi-omics approach for improving patient risk stratification and identifying targeted therapies [119]. Gulfidan et al. investigated PTC prognostic biomarkers using a meta-analysis of four transcriptome datasets, revealing key genes, proteins, metabolites, and regulatory molecules [120]. Their multi-omics data led to the identification of potential therapies that may be useful in treating PTC, including meclizine, rottlerin, and tretinoin, to name just a few. Overall, the integration of omics disciplines provides a powerful framework for revealing molecular alterations in thyroid cancer, advancing personalized treatment, and improving patient outcomes.

The most important applications of metabolomics in cancer research include distinguishing between tumor subtypes, environmental risk factors, individualized therapies, biomarker identification, and cancer pathophysiology. These are summarized in Figure 5.

## 9. Conclusions

In summary, the current application of metabolomics is limited to research studies, but there is a potential for its utilization for diagnostic purposes, treatment monitoring, and prognostication (Figure 5). However, to translate these research findings into standard clinical practice, verifying the reproducibility and preciseness of the identified biomarkers in larger diverse populations is required. Standard operating procedures for metabolomics’ wider application in thyroid cancer are needed and involve multiple steps, including standardized sample collection, processing, and streamlined analysis. It is imperative to develop models that include clinical, pathological, molecular, and metabolomics data to optimize the diagnostic procedures, discriminate between indolent and aggressive thyroid cancers, and guide individualized therapies. Developing a comprehensive framework for metabolomics can facilitate personalized treatment for thyroid cancer patients, which could lead to improved outcomes. In addition, targeting metabolic pathways is an emerging strategy to overcome resistance to the standard treatment options, and warrants further investigation in pre-clinical models and clinical trials.

## Figures and Tables

**Figure 1 cancers-17-01017-f001:**
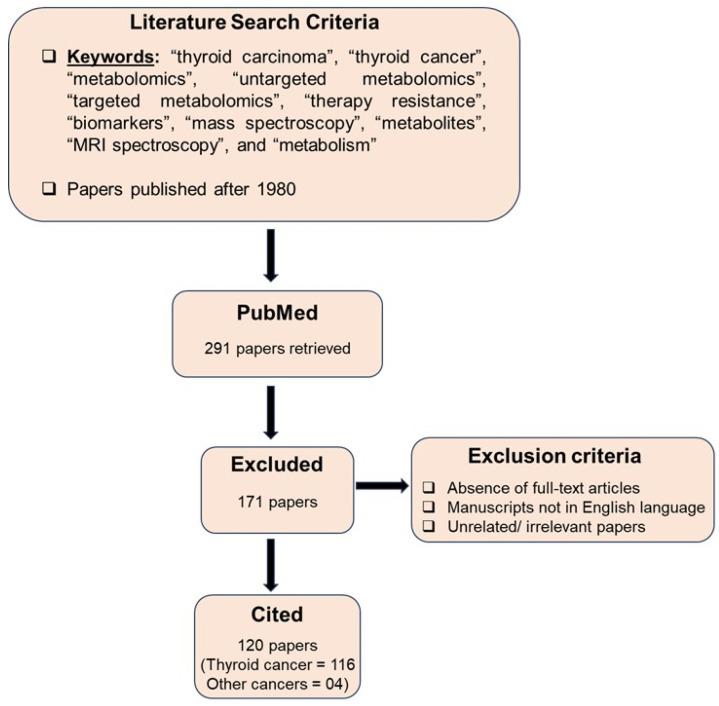
Strategy utilized for literature search and screening process.

**Figure 2 cancers-17-01017-f002:**
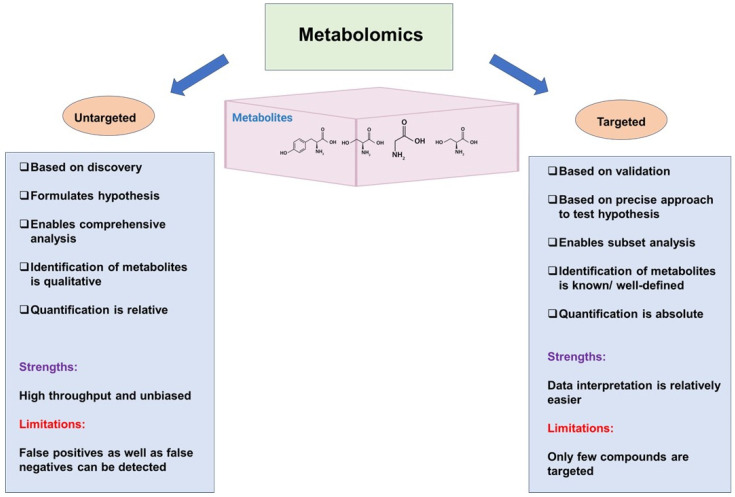
Difference between untargeted and targeted metabolomics. (BioRender software was utilized for the preparation of a portion of this figure). “Created in BioRender. Kumari, S. (2025) https://BioRender.com/n88u803, (accessed on 14 March 2025)”.

**Figure 3 cancers-17-01017-f003:**
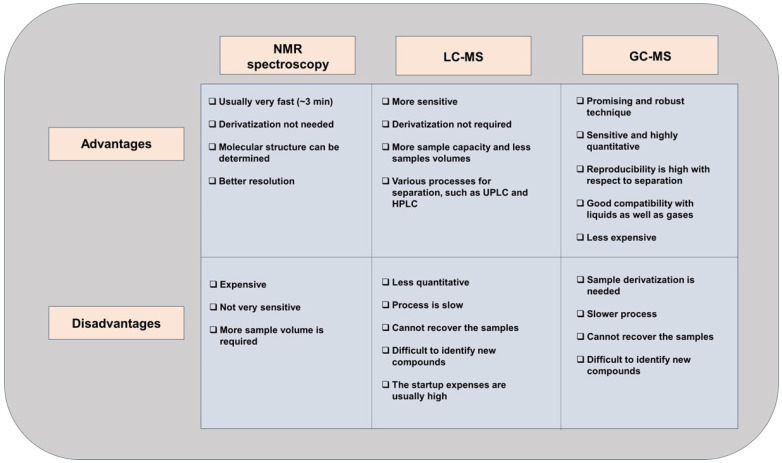
Comparison of the various analytical techniques employed for metabolomics.

**Figure 4 cancers-17-01017-f004:**
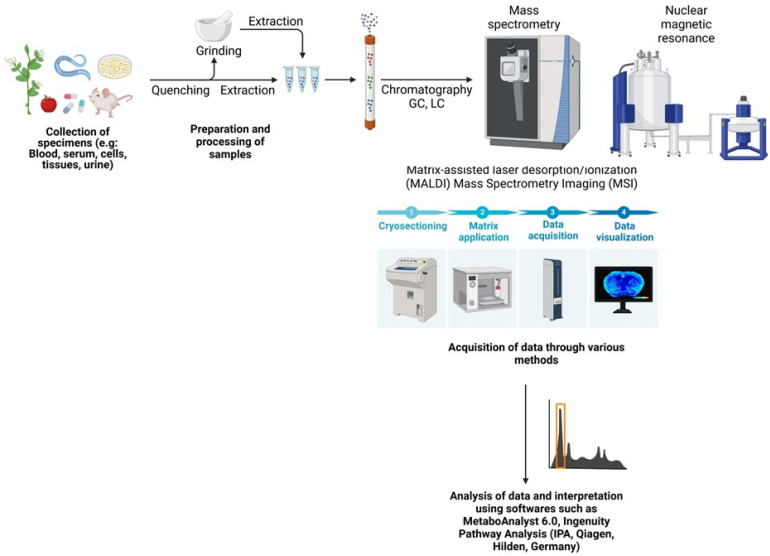
Experimental procedure for metabolomics analysis. (The figure was prepared using BioRender software). “Created in BioRender. Kumari, S. (2025) https://BioRender.com/l51c996, (accessed on 14 March 2025)”.

**Figure 5 cancers-17-01017-f005:**
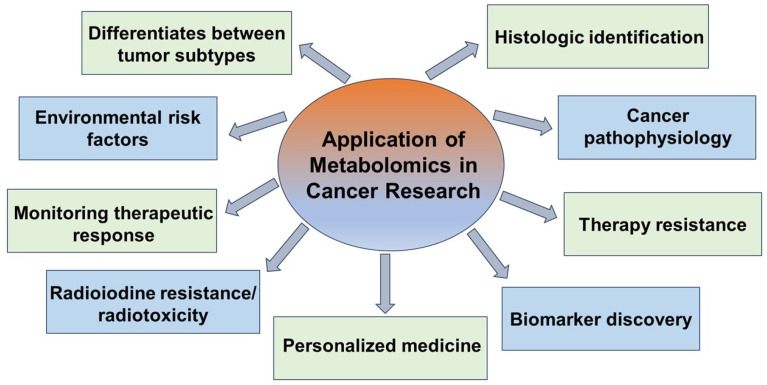
The most important applications of metabolomics in the field of cancer research.

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
