# Peer review of "Emerging Potential of Metabolomics in Thyroid Cancer—A Comprehensive Review"

_cancers, 2025, doi:10.3390/cancers17061017_

Round 1
Reviewer 1 Report
Comments and Suggestions for Authors
The Authors presented an analysis of the potential role of metabolomics in thyroid cancer detection and therapy. The article is well-written and should attract readers interested in the topic. I would suggest some minor modifications, which I have listed below.
[1] Please confirm that reference 6 (i.e., “Society, A.C., Cancer Facts & Figures 2025. . 2025”) is cited correctly. I cannot find such a reference in the literature. Please clarify.
[2] All abbreviations used in Table 1 (e.g., TC, NMR, FNA, and MS) should be explained in the footnotes.
[3] The abbreviation “PTC” is explained twice in the main text. Please do it only once the first time it is used. Similar change is needed for NIS protein. The abbreviation RAI, on the other hand, is only explained in line 343, although the abbreviation is used earlier (e.g. in lines 332 and 338).
[4] Please replace “TKI Lenvatinib” with “TKI lenvatinib.”
Author Response
Reviewer #1:
The Authors presented an analysis of the potential role of metabolomics in thyroid cancer detection and therapy. The article is well-written and should attract readers interested in the topic. I would suggest some minor modifications, which I have listed below.
[1] Please confirm that reference 6 (i.e., “Society, A.C., Cancer Facts & Figures 2025. . 2025”) is cited correctly. I cannot find such a reference in the literature. Please clarify.
Answer 1: We agree with the Reviewer’s comments and have now made appropriate changes in the reference # 6 as well as added a link to the document.
[2] All abbreviations used in Table 1 (e.g., TC, NMR, FNA, and MS) should be explained in the footnotes.
Answer 2: We appreciate the Reviewer’s suggestions and have now included the abbreviations in the footnotes of Table 1. In addition, we have included them in the “Abbreviations” list towards the end of the revised manuscript and have highlighted them in yellow.
[3] The abbreviation “PTC” is explained twice in the main text. Please do it only once the first time it is used. Similar change is needed for NIS protein. The abbreviation RAI, on the other hand, is only explained in line 343, although the abbreviation is used earlier (e.g. in lines 332 and 338).
Answer 3: We absolutely agree with the Reviewer’s comments and have now incorporated the suggested changes in our revised manuscript. The modifications are highlighted in yellow for your reference.
[4] Please replace “TKI Lenvatinib” with “TKI lenvatinib.”
Answer 4: As per Reviewer’s suggestions, we have replaced “TKI Lenvatinib” with “TKI lenvatinib.”
Reviewer 2 Report
Comments and Suggestions for Authors
Kumari and coworkers summarize in this review the potential of metabolomics in diagnosing and addressing an individualized therapy in the different types of thyroid cancer . To this end they reviewed the results of some personal experiences and of original and observational studies , reviews and meta-analysis in the field of metabolomics, published from 1980 up to 2024. From their accurate review emerges that the application of metabolomics in thyroid cancer may be useful in clarifying cancer pathophysiology, including biomarker identification and environmental risk factors, and in distinguishing between tumor subtypes and in planning tailored therapies.
COMMENT
Although many articles have been published on this topic, this review deserves some attention for its completeness in illustrating the methodological aspects in the study of metabolome and in considering also the use of this strategy in association with other methods(for example FNAB and molecular testing) to optimize diagnostic, pathophysiological and therapeutic aspects. Of particular interest the paragraph on the use of metabolomics for biomarkers discovery in cancer subtyping and that on the use of metabolomics for overcoming thyroid resistance to radioiodine therapy,which is considered a major cause of mortality in patients with thyroid cancer. Finally, the authors outline the possible link between the results of research on these aspects and the possible implications in clinical practice. I suggest that the review could be furter improved by considering in the discussion a recent experimental study by Kim YH et al( Clin Cancer Res 2024) on the integrative multi-omics analysis to reveal different metabolic phenotypes based on molecular characteristics in thyroid cancer.
Author Response
Reviewer #2:
Kumari and coworkers summarize in this review the potential of metabolomics in diagnosing and addressing an individualized therapy in the different types of thyroid cancer. To this end they reviewed the results of some personal experiences and of original and observational studies, reviews and meta-analysis in the field of metabolomics, published from 1980 up to 2024. From their accurate review emerges that the application of metabolomics in thyroid cancer may be useful in clarifying cancer pathophysiology, including biomarker identification and environmental risk factors, and in distinguishing between tumor subtypes and in planning tailored therapies.
COMMENT
Although many articles have been published on this topic, this review deserves some attention for its completeness in illustrating the methodological aspects in the study of metabolome and in considering also the use of this strategy in association with other methods (for example FNAB and molecular testing) to optimize diagnostic, pathophysiological and therapeutic aspects. Of particular interest the paragraph on the use of metabolomics for biomarkers discovery in cancer subtyping and that on the use of metabolomics for overcoming thyroid resistance to radioiodine therapy, which is considered a major cause of mortality in patients with thyroid cancer. Finally, the authors outline the possible link between the results of research on these aspects and the possible implications in clinical practice. I suggest that the review could be further improved by considering in the discussion a recent experimental study by Kim YH et al (Clin Cancer Res 2024) on the integrative multi-omics analysis to reveal different metabolic phenotypes based on molecular characteristics in thyroid cancer.
Answer: We are thankful for the Reviewer’s suggestions, and have now included the details about the Kim YH et al study (Clinical Cancer Research, 2024) under section 5 “Metabolomics in Disease Subtyping, (page 9, lines 286-290).
Reviewer 3 Report
Comments and Suggestions for Authors
The authors summarized recent reports about metabolomics research in thyroid cancer. Their review is very comprehensive and refined, encouraging readers to learn more about the potential of metabolomics as research strategy. It is giving a big hint to head and neck oncologists for future preclinical and clinical researches.
Only one minor issue should be checked. Line 321, "A study by Thakur et al. also revealed downregulated oxidative phosphorylation in response to metformin treatment [98]" needs a period.
Author Response
Reviewer #3:
The authors summarized recent reports about metabolomics research in thyroid cancer. Their review is very comprehensive and refined, encouraging readers to learn more about the potential of metabolomics as research strategy. It is giving a big hint to head and neck oncologists for future preclinical and clinical researches.
Only one minor issue should be checked. Line 321, "A study by Thakur et al. also revealed downregulated oxidative phosphorylation in response to metformin treatment [98]" needs a period.
Answer: We appreciate the Reviewer’s suggestions. As mentioned, we have now added a period after the sentence “A study by Thakur et al. also revealed downregulated oxidative phosphorylation in response to metformin treatment [99].” (page 10, lines 332-333 of the revised manuscript highlighted in yellow)